# Versatile CRISPR/Cas9 Systems for Genome Editing in *Ustilago maydis*

**DOI:** 10.3390/jof7020149

**Published:** 2021-02-18

**Authors:** Sarah-Maria Wege, Katharina Gejer, Fabienne Becker, Michael Bölker, Johannes Freitag, Björn Sandrock

**Affiliations:** 1Department of Biology, Philipps-University Marburg, 35037 Marburg, Germany; sarahmaria.wege@gmx.de (S.-M.W.); beckerfabienne@web.de (F.B.); boelker@staff.uni-marburg.de (M.B.); freitag7@uni-marburg.de (J.F.); 2Department of Physics, Philipps-University Marburg, 35037 Marburg, Germany; katharina.gejer@yahoo.de

**Keywords:** *Ustilago maydis*, CRISPR/Cas9, genetic manipulation, deletion, point mutation

## Abstract

The phytopathogenic smut fungus *Ustilago maydis* is a versatile model organism to study plant pathology, fungal genetics, and molecular cell biology. Here, we report several strategies to manipulate the genome of *U. maydis* by the CRISPR/Cas9 technology. These include targeted gene deletion via homologous recombination of short double-stranded oligonucleotides, introduction of point mutations, heterologous complementation at the genomic locus, and endogenous N-terminal tagging with the fluorescent protein mCherry. All applications are independent of a permanent selectable marker and only require transient expression of the endonuclease Cas9hf and sgRNA. The techniques presented here are likely to accelerate research in the *U. maydis* community but can also act as a template for genome editing in other important fungi.

## 1. Introduction

The biological function of the CRISPR (clustered regularly interspaced short palindromic repeats)-Cas (CRISPR-associated protein)-system was identified in archaea in 2007. It helps to acquire immunity against phages [1,2]. RNA guided recognition results in sequence-specific nucleic acid cleavage to introduce double strand breaks (DSBs) in target DNA, can be programmed and can be used outside of prokaryotes [3,4,5,6,7]. The discovery that the CRISPR/Cas system can be used for genome editing at a desired target site was awarded with the Noble prize in 2020 [8]. The CRISPR/Cas9-system and many of its variations have been used in basic research to manipulate genomes of diverse organisms [9,10,11].

The basidiomycetous fungus *Ustilago maydis* infects maize plants and has been established as a genetic model organism for studying plant infection, DNA repair, intracellular long distance transport of proteins and mRNA, secondary metabolism, and translational recoding [12,13,14,15,16,17,18,19]. Haploid *U. maydis* cells divide by budding and are highly amenable to genetic analysis. Genetic manipulation can be achieved by transformation of protoplasts using gene cassettes with antibiotic resistance markers [20,21]. *U. maydis* has a very efficient homologous recombination machinery which allows for high levels of gene replacement. This strategy is so far limited to a maximum of five independent modifications per cell by the number of available resistance markers (hygromycin, carboxin, phleomycin, gentamycin, and nourseothricin) [20,22]. In 2016, CRISPR-Cas9 editing was described for *U. maydis* to generate mutants by non-homologous end joining (NHEJ) [23]. This system was further modified for multiplex gene disruption, generation of complete deletion mutants by using 2000 bp long donor DNA for gene replacement, and by introduction of high fidelity Cas9hf enzyme to reduce potential off-target effects [24,25,26].

Here, we used the power of the CRISPR/Cas9 technology for the creation of different types of mutants in *U. maydis*. This included the generation of marker-free deletion strains but also the introduction of point mutations by site directed mutagenesis. Furthermore, CRISPR/Cas9 was used to replace a gene by its homologous gene from the related fungus *Ustilago hordei* and to generate an endogenous N-terminal fusion with the fluorescent *mCherry* marker.

## 2. Materials and Methods

### 2.1. Strains and Growth Conditions

Fungal strains were grown in liquid YEPSL (1% yeast extract (Roth, Karlsruhe, Germany), 0.4% peptone (Neogen, Lansing, MI, USA), 0.4% sucrose (Roth, Karlsruhe, Germany)) or on solid potato dextrose broth (Difco, Detroit, MI, USA) containing 1.5% Bacto agar (Roth, Karlsruhe, Germany) at 28 °C and 23 °C, respectively. The *U. maydis* strains used and generated in this study are listed in Appendix A and are derivatives of the *U. maydis* strains Bub8 [27] and MB215 [28]. *U. hordei* (Uh4857-4) was a kind gift from Regine Kahmann (MPI Marburg, Germany). To induce glycolipid production *U. maydis* strains were grown to stationary phase in YEPS-L and then transferred to nitrogen starvation medium (OD = 0.1) containing 0.17% YNB (yeast nitrogen base without ammonia, Difco, Detroit, MI, USA) and 1% glucose (Roth, Karlsruhe, Germany) as carbon source. Glycolipid production was then allowed for 6 days at 23 °C under rotation. *Escherichia coli* strain Top10 (Invitrogen, Carlsbad, CA, USA) was used for transformation according to Hanahan et al., and amplification of plasmid DNA [29].

### 2.2. Molecular Cloning and Nucleic Acid Procedures

Standard procedures were followed for generation of pSM2 and its derivatives [30,31]. Plasmids are based on the plasmid pCas9_sgRNA_0 [23]. PAM motifs were identified using www.e-crisp.org [32]. Genomic DNAs of *U. maydis* and *U. hordei* cells were prepared according to an established protocol [33]. Donor DNAs were amplified by two rounds of PCRs as illustrated in Appendix A. Primer sequences are listed in Appendix A and plasmids are listed in Appendix A. Plasmids were verified by sequencing.

### 2.3. Strain Construction

Transformation of *U. maydis* was conducted as described [34]. Briefly, 100 ng of autonomously replicating pSM2 plasmids were co-transformed with either 0.2–0.6 pmol of PCR products or 65 pmol of hybridized oligonucleotides of donor DNA. Selection was performed on carboxin (cbx, Sigma Aldrich, Buchs, Switzerland) containing plates. Transformants were verified by PCR and sequencing. Positive clones were then streaked out on cbx-free plates unless the plasmid had been lost [23].

### 2.4. Analysis of Glycolipids

Extracellular glycolipids were extracted as previously described [28]. Glycolipids (MELs and CLs) were separated by thin-layer chromatography (TLC) on silica gel 60 F_254_ plates (Merck, Darmstadt, Germany) first with a solvent system consisting of chloroform–methanol–water (65:25:4, *v*/*v*/*v*) (Roth, Karlsruhe, Germany) for 5 min followed by a second solvent system consisting of chloroform–methanol (9:1, *v*/*v*; 2 × 18 min) [35]. The plates were dried, and sugar containing compounds were visualized by application of a mixture of ethanol:sulfuric acid:p-anisaldehyde (18:1:1, *v*/*v*) (Roth, Karlsruhe, Germany; Merck, Darmstadt, Germany; Sigma Aldrich, Steinheim, Germany) followed by heating at 150 °C for 2 min [36].

### 2.5. Microscopy

*U. maydis* cells from logarithmically growing cultures were placed on agarose cushions. Cells were visualized by differential interference contrast (DIC), phase contrast (PC) and epifluorescence microscopy using a Zeiss Axiophot 200 microscope. Images were taken using a cooled Hamamatsu Orca-ER CCD camera with an exposure time of 50–300 ms. Image acquisition and deconvolution were performed using Improvision Volocity software and processing was done with ImageJ [37].

### 2.6. Kinase Inhibition

The activity of the engineered kinase Cdk5^F78G^ was inhibited by addition of the indicated amounts of NA-PP1 (Calbiochem, San Diego, CA, USA). Inhibition of Cdk5^F78G^ in logarithmically growing cells blocked cell growth immediately. The essential function of Cdk5 for normal cell growth was measured in growth assays using a micro plate reader (Synergy Mix; BioTek). 100 µl of fungal cells of OD = 0.1 were grown in YEPSL medium in the presence of NA-PP1. Growth was recorded every 30 min for 12 h in three independent experiments. Data were analyzed with Microsoft Excel software.

## 3. Results

The CRISPR/Cas system previously used in *U. maydis* is based on plasmid pCas9_sgRNA_0, which contains the gene for the Cas9 endonuclease and can be equipped with sequences encoding single guide RNA (sgRNA) for specific gene disruption in *U. maydis* [23]. Each gene disruption requires synthesis of a specific 175 bp DNA fragment for a Gibson cloning strategy [23,30]. We first modified the vector to enable direct cloning of short double stranded DNA (dsDNA) fragments containing the 19 bp target sequence of the sgRNA and some flanking sequences into pSM1 linearized with SnaBI/XbaI (Figure 1). Note that both restriction sites become inactive in the final construct. To generate plasmid pSM2, we introduced three point mutations (R661A, Q695A, and Q926A) into Cas9 to gain the high fidelity variant of Cas9, which reduces off-target cleavage [23,38].

To allow the efficient and defined deletion of genes we next determined the minimal length of donor dsDNA required for efficient homologous recombination. As a target gene we used *don3* encoding a germinal center kinase required for cell separation [39,40] because of the remarkable tree-like deletion phenotype (Figure 2A,B). To study the length of the donor dsDNA required for efficient repair of double strand breaks and introduced via Cas9, we used flanking regions of 750 bp, 250 bp, and 40 bp, respectively (Figure 2A). dsDNA was generated by two step PCR (1500 bp and 500 bp fragments containing both the left and right borders) or by hybridization of oligonucleotides (80 bp fragment) (Figure 2A). Transformants were screened for the *don3*-phenotype and further characterized by PCR (Figure 2B,C). Generation of deletion mutants was confirmed irrespective of the donor DNA applied. The use of high concentrated oligonucleotides providing only short homology was most efficient; approx. 50% of all mutants exhibiting the *don3* phenotype were deletions (Figure 2D). This strategy was also used successfully for additional genes like *rua1*, *emt1,* and *mat1* [Fabienne Becker and Björn Sandrock; unpublished data].

Chemical genetics is a powerful tool to study protein kinases *in vivo*, and is regularly used by our lab to characterize protein kinases during cytokinesis as well as for systematic mining for the function of protein kinases not yet characterized in *U. maydis,* together with undergraduate researchers [39,40,41,42,43,44]. Specific mutations in the binding domain allows for inhibition of the kinase by an external chemical. We decided to use CRISPR/Cas9 to test for the introduction of point mutations at the genomic locus, which render a kinase repressible by the ATP analog NA-PP1. As an example we decided to mutate the gene *umag_11892* encoding UmCdk5, a homolog of the cyclin dependent kinase Pho85 from *Saccharomyces cerevisiae* [45,46]. The gatekeeper in the kinase domain of UmCdk5 is located at F78 and the mutation for the chemical genetics analysis is F78G (Figure 3A). CRISPR/Cas9 dependent cleavage requires a protospacer adjacent motif(PAM) in vicinity of the cleavage site [32]. This can hamper the introduction of point mutations in regions devoid of a PAM. We searched for a PAM at a distance of around 500 bp of the codon encoding the gatekeeper amino acid [41,47]. To create donor DNA for recombination, we set up a two step-PCR with three overlapping fragments (Figure 3A and Appendix A). To avoid donor DNA disruption by Cas9 we inserted three silent mutations within the sequence corresponding to the sgRNA. For easy detection of positive transformants, a silent mutation resulting in the restriction site HindIII in the gatekeeper primers was introduced. The donor DNA was transformed together with the corresponding pSM2-Cdk5 plasmid and transformants were identified by PCR, restriction digest and sequencing (Appendix A). We succeeded in generating *cdk5* point mutations (3 of 18 transformants) and found that the kinase, which is essential in *U. maydis* [46], can be inhibited by NA-PP1 leading to an arrest of cell growth (Figure 3B).

These data demonstrate that in *U. maydis* CRISPR/Cas9 can be used to generate marker free strains with desired point mutations. In collaboration with undergraduate researchers we have already generated and tested mutated variants of the following proteins: Elm1as (L437G, UMAG_15093, 1 of 12), Mkk1as (M295G, UMAG_10855, 2 of 12), and Yak1as (L684G, UMAG_06306, 3 of 12). Strains are available upon request.

In recent years, we have characterized a gene cluster for biosynthesis of mannosylerythritol lipids (MELs) in *U. maydis* and related fungi [48,49,50]. Five similar genes are responsible for the production of these amphiphilic compounds, which differ in their acetylation and acylation patterns [51,52]. We have shown, that expression of the acetyl-transferase Mat1 from *Ustilago hordei* in *U. maydis* ∆*mat1* cells resulted in the exclusive production of mono-acetylated MEL-C molecules [50]. Therefore, we concluded that cross species complementation might be a valuable approach to customize MEL biosynthesis. For this approach marker free gene replacement would be of particular interest.

Thus, we constructed donor DNA consisting of the open reading frame (ORF) *U. hordei mat1* and 500 bp fragments from the *U. maydis mat1* gene to trigger homologous recombination (Figure 4A and Appendix A). Donor DNA (0.3 pmol) was co-transformed with pSM2-Mat1 in strain MB215 ∆*rua1*, which cannot produce the related glycolipid ustilagic acid [53]. Candidates were tested for MEL biosynthesis by growth in minimal medium without a nitrogen source. The majority of transformants produced MEL-D, the non-acetylated variant indicating disruption of the *mat1* gene. Around 5% showed the desired MEL-C production (Figure 4B). The successful exchange of the *mat1* gene was verified by PCR and sequencing (Appendix A). This strategy may also be performed for *mat1* gene from *Pseudozyma tsukubaensis* to exclusively produce MEL-B variants [54]. These findings showcase the potential of CRISPR/Cas9 for heterologous complementation in *U. maydis* in general and its potential to aid with customization of metabolic pathways.

Intracellular localization of proteins is often determined by signal sequences residing at the very N-terminus or very C-terminus of proteins [55]. While production of C-terminally tagged versions of proteins expressed from their endogenous loci is a standard procedure in *U. maydis,* N-terminal tagging usually also involves the substitution of the endogenous promoter [20,56]. Targeting of soluble proteins to cellular organelles called peroxisomes regularly depends on C-terminal extensions termed peroxisomal targeting signal type I (PTS1) [57]. Hence, PTS1 proteins might be suitable candidates to demonstrate N-terminal tagging with a fluorescent protein tag with the help of CRISPR/Cas9. Fab4 (UMAG_02182) is predicted to localize in peroxisomes and is expressed throughout the cell and the life cycle of *U. maydis* [58]. Fab4 is a putative NADPH-dependent beta-ketoacyl carrier protein reductase. It was shown that FabG, an ortholog of Fab4, performs the NADPH-dependent reduction of beta-ketoacyl-ACP substrates to beta-hydroxyacyl-ACP products, the first reductive step in the elongation cycle of fatty acid biosynthesis [59]. We engineered a donor DNA composed of four fragments fused by overlapping PCR (Figure 5A and Appendix A). Donor DNA included 1000 bp flanking sequence and the mCherry sequence upstream of the Fab4 ORF to express mCherry-Fab4. For the construction of the pSM2-Fab4 we used a PAM close to the 5′end of the ORF. Again, three silent mutations were introduced to avoid disruption of the donor DNA by Cas9 cleavage.

One positive mCherry-Fab4 expressing strain with clear peroxisomal localization was recovered from 36 candidates indicating that N-terminal tagging is possible albeit with low efficiency (Figure 5B and Appendix A). This demonstrates that in *U. maydis* N-terminal tagging under endogenous expression conditions is now possible with only one round of transformation.

## 4. Discussion

Here, we have shown different applications of the CRISPR/Cas9 system, which are likely to enhance the molecular characterization of *U. maydis*. A major limitation of the *U. maydis* system compared to *S. cerevisiae* was the requirement for very long homologous sequences to achieve deletion of genes [20,60]. This can now be compensated by the combination of CRISPR/Cas9 and highly concentrated dsDNA oligonucleotides, which can be easily generated from commercially available stocks.

Methods described here may not only be a benefit for research in *U. maydis* but may be also applicable in related fungi such as *U. hordei*, *Sporisorium reilianum,* and *Ustilago bromivora* [61,62,63]. It is possible that CRISPR may also allow for the genetic manipulation of pathogenic rust fungi, which have an enormous impact on agriculture [64].

A major limitation of the methods presented here is the strict requirement of suitable PAM sites in the vicinity to the sites of the genome for a desired genetic manipulation. While this is usually not problematic for creation of deletion mutants, other interventions such as N-terminal tagging and the creation of point mutations will strongly profit from the engineered Cas9 enzymes or alternative endonucleases [65].

Overall our study showcases different applications of CRISPR/Cas9 base genome editing, which might ultimately override many of the disadvantages researchers had, if they decided to tackle the biology of organisms outside of the traditional model systems. As multiplexing is possible with CRISPR/Cas9 systems [66,67] questions in such systems can now even be addressed systematically.

## Figures and Tables

**Figure 1 jof-07-00149-f001:**
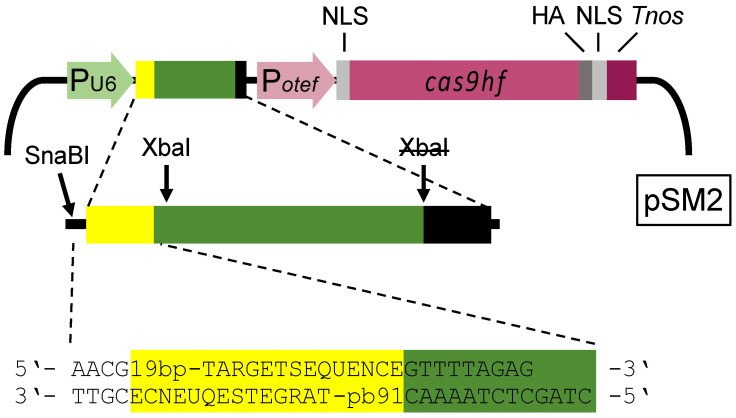
Schematic drawing of the Cas9 expression cassette and the RNA polymerase III U6 promoter (light green; P_U6_) driven sgDNA (pSM2) according to Schuster et al., 2016 [23]. Yellow: 19 bp target sequence; dark green: scaffold sequence; black: U6 terminator; light purple: constitutive otef-promoter (P_otef_); light gray: nuclear localization signal (NLS); purple: open reading frame (ORF) of high fidelity cas9; dark gray: HA-tag; dark purple: *nos* terminator. The former XbaI-site inside of the U6 terminator was inactivated by a point mutation. The 32/36 bp flanking sequences for the oligonucleotides are given.

**Figure 2 jof-07-00149-f002:**
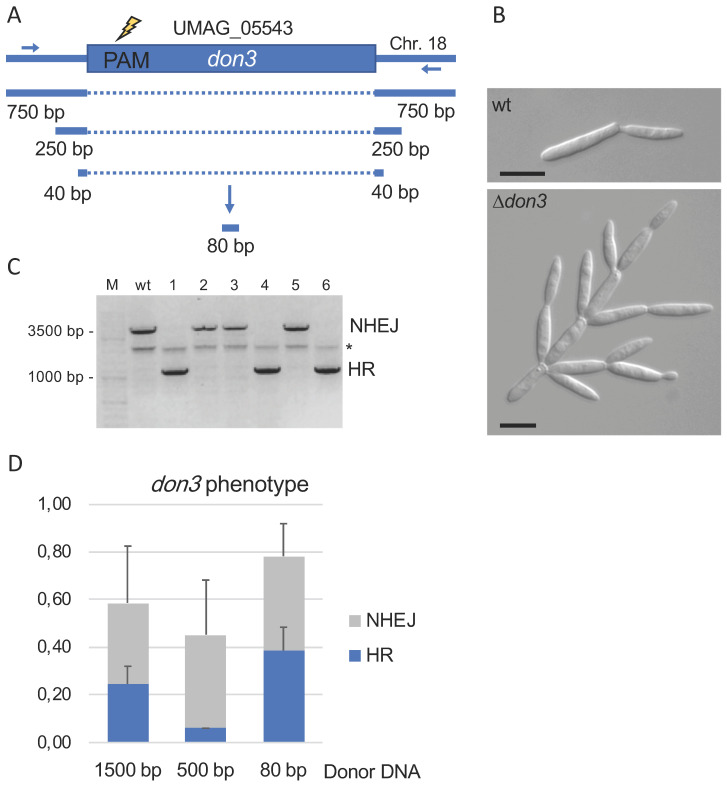
Generation of *don3* deletion mutants via donor DNA of different length. (**A**): Schematic drawing of the *don3* gene indicating the protospacer adjacent motif (PAM) in the 5′-region of the ORF, which was used. Flanking sequences of 1500 bp and 500 bp, respectively, were amplified by a two-step PCR approach. The 80 bp dsDNA was generated by hybridization of oligonucleotides. (**B**): Microscopic image (DIC) of the *don3* phenotype used for identification of deletion mutants (Scale bar 10 µm). (**C**): PCR test of six *don3* mutants either resulting from non-homologous end joining (NHEJ) or homologous recombination using donor DNA (HR). Wildtype Bub8 (wt) was used as a control. * unspecific PCR product. (**D**): Summery of three independent transformation experiments using 0.2–0.6 pmol for the PCR based donor DNAs 1500 bp and 500 bp and using 65–70 pmol hybridized oligonucleotides (80 bp).

**Figure 3 jof-07-00149-f003:**
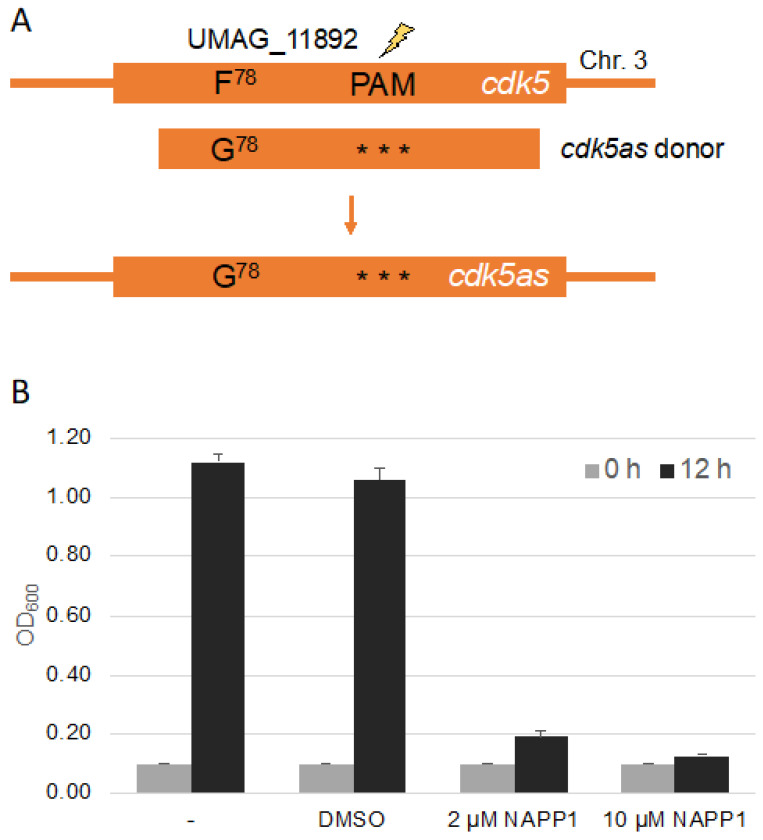
Site directed mutagenesis to generate an analog sensitive Cdk5 kinase. (**A**): Schematic drawing of the *cdk5* gene highlighting the position of the region encoding the gatekeeper and the PAM site. The donor DNA encoded the F78G mutation and contained three silent mutations within the sequence corresponding to the sgRNA. Further details are depicted in Appendix A. (**B**): Quantification of growth assays of Bub8 *cdk5as* in the absence or presence of NA-PP1 inhibitor starting with an OD_600_ of 0.1. DMSO is the solvent of NA-PP1 and served as control. Three independent experiments were analyzed.

**Figure 4 jof-07-00149-f004:**
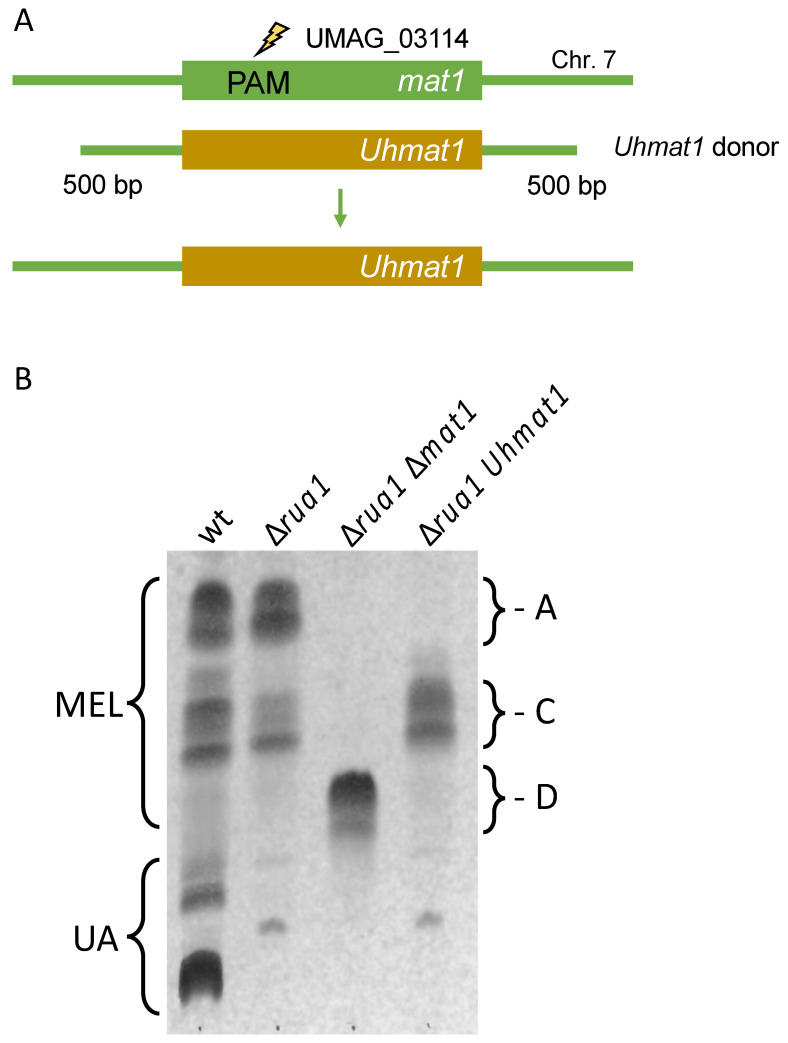
Heterologous complementation at the genomic locus. (**A**): Schematic drawing of the *U. maydis* (Um) *mat1* gene indicating the position of the PAM site. Donor DNA contained the ORF for Mat1 from *U. hordei* (Uh) and 500 bp upstream and downstream flanking sequence of the *U. maydis* ORF. Further details are depicted in Appendix A. (**B**): Thin layer chromatography of glycolipids from MB215 (wt), ∆*rua1*, ∆*rua1* ∆*mat1* and ∆*rua1* Uhmat1. MEL variants (A,C,D) can be separated according to their acetylation patterns.

**Figure 5 jof-07-00149-f005:**
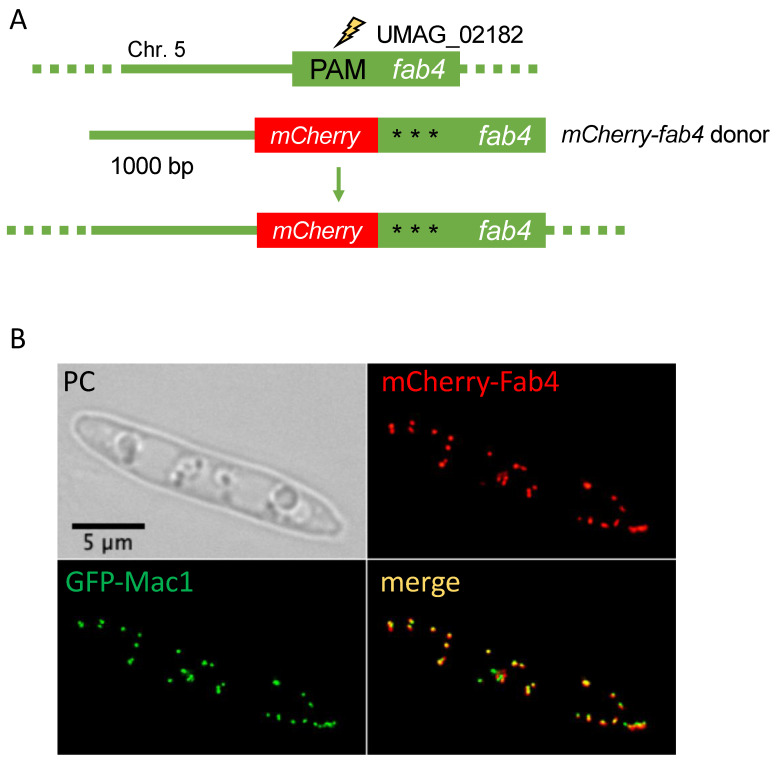
N-terminal tagging at the endogenous locus (**A**): Schematic drawing of the *fab4* gene indicating the position of the PAM site. Donor DNA consisted of 1000 bp upstream and downstream flanking sequence, the ORF for mCherry. Three silent mutations were introduced in the sequences corresponding to the sgRNA. For further details see Appendix A. (**B**): Phase contrast (PC) and fluorescent imaging of *mCherry-fab4* expressing cells. mCherry-Fab4 co-localized with GFP-Mac1 in peroxisomes [17,49]. Scale bar: 5 µm.

## Data Availability

Accession numbers from National Center for Biotechnology Information (NCBI; www.ncbi.nlm.nih.gov): *U. maydis*: Don3 (UMAG_05543; XP_011391855), Pho85 (UMAG_11892; XP_011388110), Mat1 (UMAG_03114; XP_011389465), Fab4 (UMAG_02182; XP_011388532). *U. hordei*: Mat1 (UHOR_04871; CCF52714).

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
