# Peer review of "Versatile CRISPR/Cas9 Systems for Genome Editing in Ustilago maydis"

_jof, 2021, doi:10.3390/jof7020149_

Round 1

Reviewer 1 Report

In this work, Wege and colleagues showed a catalog of Cas9/CRISPR technology applications to Ustilago maydis, which will be really useful for people researching this fungus. The main improvement showed in this work, concerning the previous one, is a much better designed sgRNA donor plasmid, which allows an easy, one-step cloning process for the sgRNA and the previously described Cas9HF allele.

The authors use four different examples to support its case: targeted deletion, point mutation, exchange of an entire ORF, and a cherry tag. All of them were successful and at an apparent handy frequency.

The major problem with this report that I have no doubt author will solve is the statistical data associated with this work:

  1. In the deletion of don3, they showed data (Fig. 2D) about the percentage of strains showing don3 phenotype and how many of these strains carried the searched substitution by the donor DNA. Are these numbers referred to in one single study? How many times was this experiment performed? Of course, the system works beautifully, but to know more precisely how efficient it is, it may be informative to choose this system versus the more classical manner to delete genes in U. maydis. Another question is why the donor's size affects the total number of colonies showing the don3 phenotype. It could make sense that the size of the donor DNA affected the ratio between NHEJ versus DSB, but what is nonsense is that the presence of don3 mutants (regardless they resulted from either NHEJ or DSB repair) was affected by the size of the donor DNA. For instance, 5/16 for 1500 bp donor, 3/16 for 500 bp donor, but 11/16 for 80 bp. These numbers were chance, or they reflect something important about the way the Cas9 acts in the presence of extrachromosomal donors? (and thereby the importance of an appropriated number of replicas).
  2. Concerning the kinase Pho85, firstly, to say that this kinase is called Cdk5 in Ustilago maydis (it was characterized more than 10 years ago), and to use an alternative gene name seems to be confusing and inelegant. Please change it accordingly. The authors showed that they successfully obtained the searched mutant, but they do not say how frequent it was. They analyzed 10 colonies, and one was mutant, or maybe they analyzed 100? Again, to have some idea about the frequency could help decide which method to use. Can they also provide the observed frequency of mutant obtention for the other kinases in which the mutation was introduced?
  3. For the substitution of an ORF or the tagging with cherry, authors provide numbers (supposedly from a single experiment): 5% in Uhmat1, 1 in 36 (2,7%) in the case of cherry:fab4. Does it mean that roughly the frequency is between 2-5% of the transformants? The tagging is marker-free (especially at the N-terminus), and this is a bonus, but it is essential to know how many colonies are expected to be positive for someone who wants to look for mutants.

In summary, the tool is excellent. Therefore, it merits publication, but having a little more information about frequencies will improve the work and attract more people to use this system.

Author Response

The major problem with this report that I have no doubt author will solve is the statistical data associated with this work:

1. In the deletion of don3, they showed data (Fig. 2D) about the percentage of strains showing don3 phenotype and how many of these strains carried the searched substitution by the donor DNA. Are these numbers referred to in one single study? How many times was this experiment performed? Of course, the system works beautifully, but to know more precisely how efficient it is, it may be informative to choose this system versus the more classical manner to delete genes in U. maydis. Another question is why the donor's size affects the total number of colonies showing the don3 phenotype. It could make sense that the size of the donor DNA affected the ratio between NHEJ versus DSB, but what is nonsense is that the presence of don3 mutants (regardless they resulted from either NHEJ or DSB repair) was affected by the size of the donor DNA. For instance, 5/16 for 1500 bp donor, 3/16 for 500 bp donor, but 11/16 for 80 bp. These numbers were chance, or they reflect something important about the way the Cas9 acts in the presence of extrachromosomal donors? (and thereby the importance of an appropriated number of replicas).

Our comment:

We repeated the experiment two times to improve the data (see novel Fig. 1D). Please note that high efficiency of small fragments to create deletions is probably the result of the higher DNA concentrations obtained by annealing of oligonucleotides compared to PCR. We did not test lower concentrations of oligonucleotides or higher concentrations of longer DNA fragments, as using annealed oligonucleotides might anyway be the most convenient approach.

2. Concerning the kinase Pho85, firstly, to say that this kinase is called Cdk5 in Ustilago maydis (it was characterized more than 10 years ago), and to use an alternative gene name seems to be confusing and inelegant. Please change it accordingly.

Our note:

We appreciate this point and modified the manuscript accordingly.

The authors showed that they successfully obtained the searched mutant, but they do not say how frequent it was. They analyzed 10 colonies, and one was mutant, or maybe they analyzed 100? Again, to have some idea about the frequency could help decide which method to use. Can they also provide the observed frequency of mutant obtention for the other kinases in which the mutation was introduced?

Our comment:

We now provide numbers for targeting efficiency for cdk5 and the other genes. We usually analyzed around twelve colonies and found comparable efficiencies (see text). The efficiency of targeting may vary substantially depending on the PAM site and the sequence context. We routinely use e-crisp.org for prediction of putative target sites and found it reliable.  

3. For the substitution of an ORF or the tagging with cherry, authors provide numbers (supposedly from a single experiment): 5% in Uhmat1, 1 in 36 (2,7%) in the case of cherry:fab4. Does it mean that roughly the frequency is between 2-5% of the transformants? The tagging is marker-free (especially at the N-terminus), and this is a bonus, but it is essential to know how many colonies are expected to be positive for someone who wants to look for mutants.

Our comment:

Indeed, tagging of fab4 was not very efficient as also stated in the manuscript. However, we think it is worth to try if there is an application. It is so far the most convenient way to make changes at the N-terminus of a protein and to keep the promoter intact at the same time. It will of course also depend on the quality of the PAM site in vicinity to the N-terminus. We discuss the limitation of the system in more details but also highlight the potential solutions via engineered versions of Cas9 or other Cas enzymes.

In summary, the tool is excellent. Therefore, it merits publication, but having a little more information about frequencies will improve the work and attract more people to use this system.

            Thank you.

Reviewer 2 Report

Dear authors,

Thank you for the nice paper about the improved usage of CRISPR in Ustilago.

I only have some small suggestions:

-Could you please include some/few sentences about PAM as it is such a limiting factor.

-Fig 1D: phenotype and deletion: are those the frequencies of observed phenotypic and genetic confirmed mutants? If yes, describe it in the legend as such.

-Fig. 5B: Would it be possible to include a peroxisome staining picture (with a dye or a marker protein labelled with a different fluorophore)?

-line 24 include “of” after outside

-line 145 include “of” after customization

-Funding: move the funding section here (from the acknowledgments)

-Acknowledgements:

Include “are” after we at line 280

Author Response

Thank you for the nice paper about the improved usage of CRISPR in Ustilago.

Thank you!

 I only have some small suggestions:

-Could you please include some/few sentences about PAM as it is such a limiting factor

We discuss the limitation of the tool due to the PAM site in the discussion section. We also refer to possible solutions.

-Fig 1D: phenotype and deletion: are those the frequencies of observed phenotypic and genetic confirmed mutants? If yes, describe it in the legend as such.

Deletion refers to verified deletion mutants. We now included two additional replicates of the experiment and did verify all mutants by PCR. The other mutants will probably contain smaller deletions and insertions but did not undergo homologous recombination (see new figure 1D).

-Fig. 5B: Would it be possible to include a peroxisome staining picture (with a dye or a marker protein labelled with a different fluorophore)?

We have generated a strain also expressing the peroxisomal acyltransferase GFP-Mac1 and provide evidence for colocalization.

-line 24 include “of” after outside

-line 145 include “of” after customization

 -Funding: move the funding section here (from the acknowledgments)

 -Acknowledgements:

Include “are” after we at line 280

We have addressed these points.

Reviewer 3 Report

In this study the authors present an optimized method for rapid and reliable genetic manipulation of Ustilago maydis using a modified version of the Cas9 nuclease and oligonucleotides/PCR amplificates as donor DNAs. These findings allow to perform e.g. site-directed mutagenesis and gene-fusions in U. maydis without the tedious use of marker cassettes while keeping the native genetic environment intact. The study is well-written and certainly of interest to the community. I only have some minor comments as outlined below.

I am a bit confused about the authors strategy to vary the target sequence to avoid disruption of the donor DNA and a second cutting event on the DNA. I suppose the authors mixed the term “PAM” with the whole Cas9 target sequence in Fig. S1. The PAM is only the three amino acid NGG motif adjacent to the target sequence. Furthermore, the authors exchanged a “A” to “G” in the PAM sequence. Maybe I’m mistaken but in my opinion this does not destroy PAM sequence. I would be grateful if the authors could comment on how they chose the point mutations within the target sequence. Furthermore, the respective bases could be indicated in the sequencing image of Fig. S1.

To generate don3 deletion mutants, the authors give the individual amounts of donor DNA used during transformation. It would certainly be interesting to see if using larger amounts of the PCR-amplified donor DNAs increase the yield of positive transformants, however I see that this would be beyond the scope of the manuscript. Still, I would be grateful if the authors could add the amount of donor DNA used for the generation of pho85as, Uhmat1 and mCherry-fab4.

The funding section needs to be changed. The authors should either remove it or acknowledge their funding through the German Research Foundation (which is currently under acknowledgements).

Line 104: Consider changing to “…three silent mutations within the sequence corresponding to the sgRNA.” 

Author Response

I am a bit confused about the authors strategy to vary the target sequence to avoid disruption of the donor DNA and a second cutting event on the DNA. I suppose the authors mixed the term “PAM” with the whole Cas9 target sequence in Fig. S1. The PAM is only the three amino acid NGG motif adjacent to the target sequence. Furthermore, the authors exchanged a “A” to “G” in the PAM sequence. Maybe I’m mistaken but in my opinion this does not destroy PAM sequence. I would be grateful if the authors could comment on how they chose the point mutations within the target sequence. Furthermore, the respective bases could be indicated in the sequencing image of Fig. S1.

Sorry for this confusion. We did not mutate the PAM sequence. In Fig. S1 the terms 5’ PAM and 3’ PAM only referred to our names of the particular oligonucleotides used to generate the repair DNA fragment. We introduced three silent changes within the nucleotides pairing with the sgRNA to prevent cleavage after integration but to keep the amino acid sequence intact. To a better understanding we have changed the names to 5’ SGmut and 3’ SGmut (also in Fig S3). Mutations to modify the binding pocket of the kinase and the sgRNA target are depicted in Fig. S1B.

To generate don3 deletion mutants, the authors give the individual amounts of donor DNA used during transformation. It would certainly be interesting to see if using larger amounts of the PCR-amplified donor DNAs increase the yield of positive transformants, however I see that this would be beyond the scope of the manuscript. Still, I would be grateful if the authors could add the amount of donor DNA used for the generation of pho85as, Uhmat1 and mCherry-fab4.

The amounts of donor DNAs used were added at least in the methods section.

The funding section needs to be changed. The authors should either remove it or acknowledge their funding through the German Research Foundation (which is currently under acknowledgements).

Line 104: Consider changing to “…three silent mutations within the sequence corresponding to the sgRNA.” 

The other requested changes were made.

Best regards

Round 2

Reviewer 1 Report

All my requests were answered. I want to thank the authors for the willingness to address all queries